# A deep learning-based pipeline for error detection and quality control of brain MRI segmentation results

**Irene Brusini**[1,2]                                                          BRUSINI@KTH.SE
**Daniel Ferreira Padilla**[2]                              DANIEL.FERREIRA.PADILLA@KI.SE
**José Barroso**[3]                                                      JBARROSO@ULL.EDU.ES
**Ingmar Skoog**[4]                                              INGMAR.SKOOG@NEURO.GU.SE
**Örjan Smedby**[1]                                                          ORSME@KTH.SE
**Eric Westman**[2]                                                  ERIC.WESTMAN@KI.SE
**Chunliang Wang**[1]                                                      CHUNWAN@KTH.SE

[1] *Department of Biomedical Engineering and Health Systems, KTH Royal Institute of Technology, 141 52 Huddinge, Sweden*

[2] *Department of Neurobiology, Care Sciences and Society, Karolinska Institute, 141 83 Huddinge, Sweden*

[3] *Department of Clinical Pshychology, Psychobiology and Methodology, Universidad de La Laguna, 38207 La Laguna, Spain*

[4] *Department of Psychiatry and Neurochemistry, Sahlgrenska Academy at the University of Gothenburg, 413 45 Gothenburg, Sweden*

## Abstract

Brain MRI segmentation results should always undergo a quality control (QC) process, since automatic segmentation tools can be prone to errors. In this work, we propose two deep learning-based architectures for performing QC automatically. First, we used generative adversarial networks for creating error maps that highlight the locations of segmentation errors. Subsequently, a 3D convolutional neural network was implemented to predict segmentation quality. The present pipeline was shown to achieve promising results and, in particular, high sensitivity in both tasks.

**Keywords:** brain MRI, segmentation, quality control, GANs, 3D CNN

## 1. Introduction

Brain MRI segmentations can be affected by errors and their quality control (QC) is usually carried out visually, which can be very subjective and time consuming. Previous studies have proposed deep learning-based methods for performing automatic QC on medical image segmentation results (Robinson et al., 2018; Valindria et al., 2017; Galdran et al., 2018; Roy et al., 2018). However, these solutions do not point out the errors' locations, which would be useful to understand their impact on research results and speed up their correction. In this work, we propose a method that automatically (1) locates segmentation errors by employing conditional generative adversarial networks (cGANs), (2) classifies segmentation quality by using a convolutional neural network (CNN).

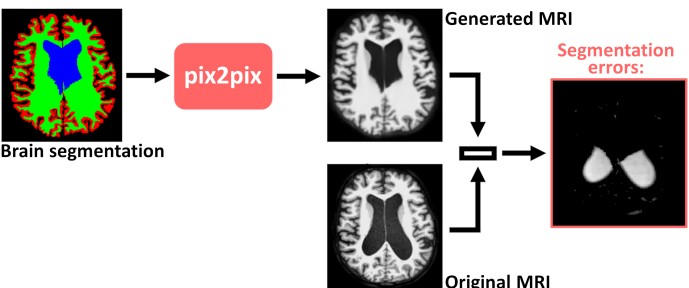

Figure 1: Schematic representation of the method for generating error maps. It is shown for the axial view, but the same idea is applied on all three views.

## 2. Methods

Given a series of input MRI images $I$ and their segmentations $S$, our goal is to learn a mapping from a certain $S$ to the original MRI $I$. For this purpose, a pix2pix model (Isola et al., 2017) was defined for each image view (axial, coronal and sagittal). Each model receives as input a segmentation slice indicating gray matter (GM), white matter (WM) and cerebrospinal fluid (CSF). These segmentations were generated with FreeSurfer 6.0 (Fischl, 2012). By contrast, the output of each pix2pix model is an MRI slice that is expected to match the input segmentation. The generated and the original MRI slice can be compared after intensity normalization. Their difference highlights the regions where they do not match, i.e. where segmentation errors can be present (see Figure 1). The difference images from all slices and all views can then be aggregated together in a 3D error map. These maps are finally refined with post-processing steps, including thresholding and Gaussian smoothing.

The generated error map and the original MRI image are fed as input into a 3D CNN to predict if the segmentation is good (output 0) or bad (output 1). The CNN was modeled using six convolutional layers (with $3 \times 3 \times 3$ kernels and, respectively, 32, 32, 64, 64, 128 and 128 units) intermingled with batch normalization layers. These are finally followed by a global average pooling layer and two dense layers (with, respectively, 128 and 1 unit).

We used images of size $256 \times 256 \times 256$ mm$^3$ from two diagnostic groups (healthy and Alzheimer's patients) and three cohorts: ADNI (Jack et al., 2008), GENIC (Machado et al., 2018) and H70 (Rydberg Sterner et al., 2019). We selected 1600 subjects whose segmentations had been visually rated as accurate. From each of these subjects, 10% of the segmented image slices (from all views) were randomly included in the training set of the cGANs. The model was then tested on other 600 subjects having segmentation errors and 190 subjects with accurate segmentations. Subsequently, the error maps and original MRI images were down-sampled to half resolution to be used as inputs for the classification CNN. A class-balanced dataset was obtained by randomly selecting 300 subjects having accurate segmentations and 300 presenting errors. The performance of the CNN was investigated using 10-fold cross-validation on these balanced data.

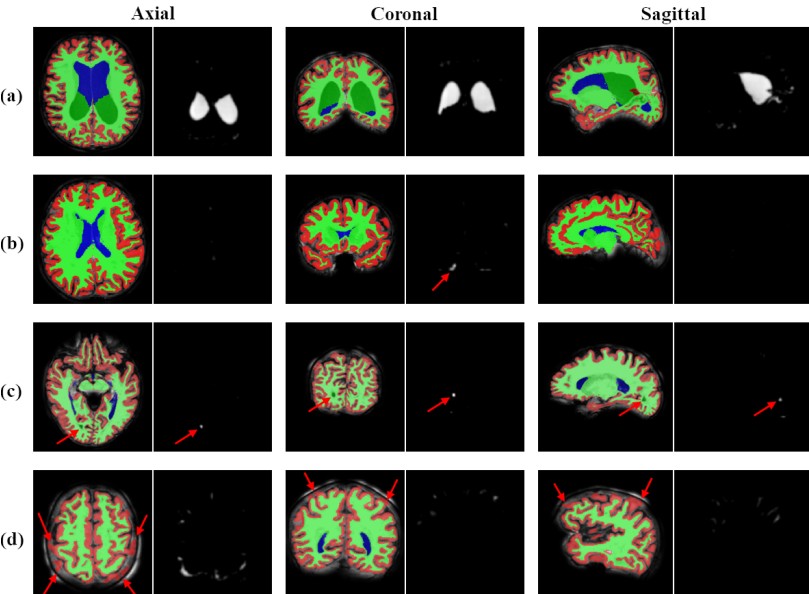

Figure 2: Examples of slices of 3D error maps. For each view, the segmentation result (red = GM, green = WM, blue = CSF) and its error map are shown. In (a), a large segmentation error was accurately detected. In (b), a false positive (caused by subject-specific anatomical features) was found in the error map (red arrow). In (c), a small CSF misclassification (red arrows) was identified on a segmentation that had been visually rated as good. In (d), the method fails at highlighting part of several cortical overestimations (red arrows).

## 3. Results

The cGAN-based method was generally effective in locating segmentation errors. It successfully detected particularly large errors in Alzheimer's brains, whose segmentations are usually more challenging because of their neurodegenerative patterns (Figure 2a). The effectiveness of the method was also evident in some cases where small errors were identified in segmentations that had been rated as accurate with visual QC (Figure 2c). On the other hand, a few false positives (i.e. highlighted regions that do not actually correspond to errors) were identified in most subjects (Figure 2b). Moreover, the error maps are based on intensity differences, so errors in regions with high contrast (e.g. WM vs. CSF) were better highlighted than those in areas having similar intensity (e.g. cortex vs. meninges), as shown in Figure 2d.

To investigate the performance of the classification CNN, we computed the area under the ROC curve, which was equal to 0.85. By setting a classification threshold of 0.4, the highest accuracy (i.e. 0.80) was obtained, with a recall of 0.83 and a precision of 0.77. By lowering the threshold to 0.3, a high recall of 0.96 was observed against a precision of 0.56.

## 4. Conclusions and future work

The proposed deep learning-based pipeline was shown to be promising for both automatic QC and localization of brain MRI segmentation errors with high sensitivity. This tool

could thus be used not only to check segmentation quality, but also to speed up the error correction and to evaluate the reliability of segmentation results in certain brain regions.

One limitation is the high presence of false positives (segmentations wrongly classified as bad) in both the error maps and the quality classification. We believe that a high sensitivity to bad segmentations is yet preferable to a high specificity, because it limits the use of wrong segmentation results in research studies. However, we still aim at increasing the precision by extending the post-processing steps on the error maps and testing other networks for the quality classification. Moreover, for comparing the generated and original MRI slice, we will test other techniques that are less intensity-dependent.

## Acknowledgments

This project is financially supported by the Swedish Foundation for Strategic Research (SSF), the Swedish Research council (VR), the joint research funds of KTH Royal Institute of Technology and Stockholm County Council (HMT), the regional agreement on medical training and clinical research (ALF) between Stockholm County Council and Karolinska Institutet, the Swedish Alzheimer foundation and the Swedish Brain foundation.

Data collection and sharing for this project was funded by the Alzheimer's Disease Neuroimaging Initiative (ADNI) (National Institutes of Health Grant U01 AG024904) and DOD ADNI (Department of Defense award number W81XWH-12-2-0012). ADNI is funded by the National Institute on Aging, the National Institute of Biomedical Imaging and Bioengineering, and through generous contributions from the following: AbbVie, Alzheimer's Association; Alzheimer's Drug Discovery Foundation; Araclon Biotech; BioClinica, Inc.; Biogen; Bristol-Myers Squibb Company; CereSpir, Inc.; Cogstate; Eisai Inc.; Elan Pharmaceuticals, Inc.; Eli Lilly and Company; EuroImmun; F. Hoffmann-La Roche Ltd and its affiliated company Genentech, Inc.; Fujirebio; GE Healthcare; IXICO Ltd.; Janssen Alzheimer Immunotherapy Research & Development, LLC.; Johnson & Johnson Pharmaceutical Research & Development LLC.; Lumosity; Lundbeck; Merck & Co., Inc.; Meso Scale Diagnostics, LLC.; NeuroRx Research; Neurotrack Technologies; Novartis Pharmaceuticals Corporation; Pfizer Inc.; Piramal Imaging; Servier; Takeda Pharmaceutical Company; and Transition Therapeutics. The Canadian Institutes of Health Research is providing funds to support ADNI clinical sites in Canada. Private sector contributions are facilitated by the Foundation for the National Institutes of Health (www.fnih.org). The grantee organization is the Northern California Institute for Research and Education, and the study is coordinated by the Alzheimer's Therapeutic Research Institute at the University of Southern California. ADNI data are disseminated by the Laboratory for Neuro Imaging at the University of Southern California.

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
