# OpenReview forum: "A deep learning-based pipeline for error detection and quality control of brain MRI segmentation results"
_MIDL.io/2020/Conference — MIDL 2020_

### Official Review · AnonReviewer1 · 2020-03-11
**Novel approach, nice initial validation.**

**Rating:** 4
**Confidence:** 4

**Review:**

This work presents a deep learning approach for error detection of automated segmentation pipelines. The model uses the previously published pix2pix conditional GAN model to learn the original image from the segmentation. In the test phase the predicted image is compared to the original image using a CNN, and an error map is generated.
Novel approach, nice initial validation. The methods shows good performance, but some false positives that should still be addressed.

---

### Official Review · AnonReviewer3 · 2020-03-11
**Interesting approach for detection of segmentation errors**

**Rating:** 3
**Confidence:** 4

**Review:**

This short paper proposes a method for detection of segmentation errors. First, a network (cGAN) is trained to predict the original image based on its segmentation. Differences between the predicted image and the original image are an indication of segmentation errors. A second network takes this difference image as input, together with the original image, and predicts whether the segmentation was acceptable or not. Evaluation results are promising.
Strengths:
- Relevant topic
- Method seems original
- Promising results.
Weaknesses:
- Some motivation for the method is lacking. Why not directly train a classifier that takes the segmentation and the original image as input, and predicts whether the segmentation was acceptable or not? Why do we need to first predict the original image? Experimental comparison to such simpler approach would have made the paper stronger.
- The experimental setup is a bit unclear. Specifically, it is not clear whether the final class-balanced dataset of 300+300 subjects had overlap with the previously mentioned datasets of 1600/600/190 subjects. This would be suspicious, since the cGAN was trained on part of that data.

---

### Official Review · AnonReviewer2 · 2020-03-13
**MRI segmentation quality control that provides useful localization error via GAN and possibly less meaningful binary good/bad classification via CNN**

**Rating:** 2
**Confidence:** 4

**Review:**

The authors propose a method for brain MRI segmentation quality control (QC). Their method makes use of pix2pix to generate a synthetic MRI from the segmentation result and then compares the synthetic and original MRIs to create an error map. This error map and the original MRI are then input to a CNN that classifies the result as either good or bad.

Strengths:
-	Using pix2pix to generate synthetic MRI in order to create error maps is interesting and can be used to localize areas of segmentation error, and initial results appear promising.
-	Training (n=1,600 subjects) and testing (n=~800) set sizes are large, demonstrating robust evaluation.

Weaknesses:
-	Evaluation of the segmentation error maps is limited to qualitative visual inspection. Some form of qualitative evaluation would be useful.
-	It is unclear how the training/testing images were scored and under what criteria, which would be useful for understanding the good/bad rating system.
-	While segmentation QC is a valuable tool, summarizing the entire segmentation quality into a single binary good/bad may not be useful for practical use. It is very subjective to say that something is good/bad, for example Fig 2a and Fig 2c show dramatically different severity of segmentation error. This “amount” of error may be more valuable than binary good/bad, and let the users decide their tolerance for error.

This is an interesting topic, and I generally like the pix2pix approach to compare against the original imaging as a way to get to a segmentation error map; however, I am less enthusiastic about second half of the proposal, the classification methodology, as I think binary evaluation does not necessarily have straightforward clinical utility. Instead, it seems that quantification of the error map as a measure of quality would be more useful.

---

### Official Review · AnonReviewer4 · 2020-03-16
**This work is in its preliminary stages and practical implementation in neuroimaging studies not realistic**

**Rating:** 1
**Confidence:** 4

**Review:**

Quality: mediocre - while there is an interesting idea regarding comparing simulated MRI data to real scans to detect errors in segmentation the practical implementation in neuroimaging studies not realistic with the FPR shown in the results

Clarity: The paper is clearly written, but there is a lack of details in the data description. While three studies are mentioned it is not clear how exactly training and test data were divided and also parameter selection is not described at all.

Originality: I am not completely familiar with all the literature they cite but it seems that their idea comparing simulated MRI data to real scans to detect errors in segmentation is novel - but this might also have been presented at another conference before. At least this is not widely known.

Significance: While a solution for QC of brain segmentations is a very important problem, th paper does not present any significant practical solution, hence it's significance is low.

Pros:
- the need for good QC tools in MRI neuroimaging is of high significance
- the idea to detect segmentation errors from differences between synthesized and real MRI seems to be novel
Cons:
- practical implementation in neuroimaging studies not realistic
- as an exhaustive freesurfer user, I am aware that sensitivity/recall not the only thing that's important, but rather the false positive rate or precision; for every chnage in a segmentation FreeSurfer needs to be run again, so a high rate of false positives will make this procedure not practical compared to visual QC
- the errors shown in Figure 2 d) that cannot be fixed with the method are some of the most crucial errors when doing cortical segmentation with FreeSurfer
- the proposed method uses FreeSurfer, but doesn't utilize the real power of FreeSurfer by using the surface parcellations; in principle it does not utilize FreeSufer to its full extent if the cortical ribbon quality is judged by segmentations instead of surface parcellations
- the way the training and test data are described is not clear
- the description of the parameter selection is lacking

---

### Meta-Review · Area_Chair1 · 2020-04-06
**MetaReview of Paper198 by AreaChair1**

**Rating:** 3

**Metareview:**

The question of segmentation QC is very important.
However, some aspects of the validation protocol remains questionable (e.g. correct separation between testing/training).

**Paper Type:**

validation/application paper

---

### Decision · Program_Chairs · 2020-04-11

Accept